# OpenReview forum: "Endless Jailbreaks with Bijection Learning"
_ICLR.cc/2025/Conference — ICLR 2025 Poster_

### Official Review · Reviewer_sNn1 · 2024-11-01

**Soundness:** 3
**Presentation:** 3
**Contribution:** 3
**Rating:** 8
**Confidence:** 4

**Summary:**

This paper introduces an adversarial attack called "bijection learning" that exploits the in-context capabilities of large language models (LLMs) by teaching it a simple cipher to bypass safety guardrails. By teaching LLMs an arbitrary string-string mapping, the attack encodes harmful queries in a "bijection language" which can easily be decoded back to English. The approach is effective across various models, with the attack's potency increasing with model capabilities. The authors demonstrate this with extensive experiments, showing high Attack Success Rates (ASR) on a range of safety-trained frontier models such as Claude and GPT-4o. The paper argues that as models become more capable, they become more susceptible to cipher attacks like bijection learning, posing significant safety challenges.

**Strengths:**

- **Simple and scalable attack that exploits in-context learning in LLMs**: The bijection learning attack method, allows you to sample a large number of text mappings that transform the input text into a bijection language. This allows you to sample them until you find a successful attack, which is a powerful red-teaming scheme.
- **High ASR on frontier models**: The attack achieves an ASR of 94% on Claude 3 Opus, which is impressive.
- **In-depth analysis of how effective the attack is with different scales**: The authors find that the attacks are stronger with scale. Smaller models fail to learn bijections, but the attack can be tuned for difficulty by changing the “fixed size” to work on less capable LLMs.
- **Contributions to Safety Research**: The paper identifies a new risk factor in AI safety that scales with model capabilities, emphasising the dual-use nature of advanced reasoning in LLMs. It underscores the necessity for evolving safety mechanisms that match these capabilities, providing crucial insights for AI developers on robust mitigation strategies.

**Weaknesses:**

- **Claims need to be better explained or backed up with a citation:**
    - “Scale of models and safety is under explored”. I’m not sure this is true because most jailbreak attack papers text over various scales. MSJ, Jailbroken, etc, all look at this (or they at least look at how safety changes with scale). You need to make this claim more specific because I agree that using the improved capability to attack via ciphers is underexplored.
    - “Model capability increasing also widens the attack surface”. I think this is unclear. It does introduce new novel vulnerabilities, but it could also be the case that the total attack surface shrinks even when there are these new cipher-based vulnerabilities. So again, I would make this claim more specific and less general (unless you have data to back it up or can cite a paper that does show this generally).
    - Is your method for jailbreaking novel? Ciphers are not new, but perhaps your version of bijection learning in context is novel. I think it is worth being clearer on what is novel and also doing a better job at comparing and contrasting in the related work section.
    - Is the approach scale agnostic? Perhaps to a certain point but this would break down? What is the smallest LLM you tried the approach on? You say “new language model risk factors may emerge with scale” but also say the approach is “scale agnostic”. I think making the story clearer and not mixing the two here would be good. Also, it is important not to confuse bigger models with improved alignment fine-tuning, and it can be tricky to make claims about this since labs do not release details of their training.
    - “while many other jailbreak methods only work on smaller models” - can you cite this or explain more? Most jailbreak techniques, e.g., GCG, PAIR, TAP, etc, work well across all model sizes, so I am not sure this claim is true.
    - “more powerful on more capable models” - can you quantify this?
    - “arguable the safest frontier model at time of writing” - can you cite this result or remove it? Gemini 1.5 Pro is very competitive and I’m not sure which is ultimately better.
    - “endless mappings that yield successful jailbreaks” - how many templates did you test, and how many of them worked? It would be good to quantify this in your contributions.
    - “certain complexities of bijection learning systematically produce certain failure models, giving further insight into our scaling laws” - what is the complexity of bijection learning? What are the failure modes? Can you give a one-sentence summary to help ground this claim?
    - “more severe harmful behaviors more frequently” - how much more? Can you give an idea of the jump from 3.5-level models to 4? How do you know that it is bijection learning that induces more harmful behaviour? It could simply be because the model is more capable. Perhaps compare “egregiousness” with other jailbreaks and see if bijection learning induces more harmful ones. Otherwise, this claim isn’t interesting.
    - Say more about your scaling laws in the contribution section - do they allow you to forecast future capabilities? What equation are you fitting?
    - “harmful model responses produced by our attack are thus fully reproducible” - have you tested this on all frontier models? Even at temp 0, output diversity exists, so be careful of your claim here.
    - “Our work is the first to produce attacks that are simultaneously black-box, automated, and scale agnostic” - I don’t think this is the case. PAIR and TAP are prime examples of methods that fit these criteria.
    - In the results section: “[whitebox methods] fail to transfer to the newest” - could you cite this? There is some GCG transfer. If you have numbers, then include them in the paper (even if it is in the Appendix)
- **Some discussion points can be improved:**
    - I think you can motivate your work better e.g. many attacks are caught by defenses such as output classifiers, but your work can bypass these easily by using a cipher that the output classifier won’t understand. However, it is unclear if you use an input+output classifier if they will catch your attacks or not. The classifier must be at the same capability level as the generation model.
    - The discussion of the “bijection jailbreaks can be universal” can be improved by making the point that the algorithm itself leads to a universal attack and relies on a “fuzzing” technique that samples a prompt template until one works. See LLM fuzzer paper https://www.usenix.org/conference/usenixsecurity24/presentation/yu-jiahao.
- **Related work lacks comparing and contrasting with their work:**
    - Safety of LLMs - this does not compare and contrast with your work. Is it even relevant
    - Adversarial attacks against LLMs - please compare and contrast more. Maybe separate the cipher work into its own section and contrast it in more fine-grain detail.
    - Adversarial robustness in LLMs - is this relevant to your work since you don’t look into defenses?
- **Lack of threat model:** Why are you working on black-box attacks on frontier models? Why is this more impactful to work on than white-box attacks? (I think it is)
- **Improving method explanation**:
    - Desiderata section. Universal and/or automated - this sentence is hard to parse, perhaps separate into two bullets
    - Fixed size (section 2.2) - this bullet point makes it hard to understand what you mean. It is a simple concept, and I think it could be explained more clearly and with fewer words. I think it would be easier to define complexity, C, as the number of characters that map to something else. Then it is easy to understand that C=0 means there is no change, and C=26 means you change each letter for something else.
    - The false positive rate is not measured. Report the false positive rate when you talk about it. Also, do you check every single response in all your experiments? Some more clarity on your method here (including a rubric your human evaluators follow) would be good as it impacts the ASRs throughout the paper a lot and will help people replicate.
    - Add how you filter HarmBench, e.g. do you just use standard direct requests and remove copyright/context ones?
- **Lack of black-box baselines and explanation of baselines used**:
    - There should be more baselines, e.g. vs PAIR and TAP. You could evaluate these with the BoN methodology too. I expect PAIR to get similar ASRs on GPT4o-mini. Without these, it makes
    - Please explain your implementation for ascii, caeser-cipher, morse code and self-cipher. (I suggest significantly cutting down section 5 to make room). The reader is left guessing how these work. Is it fair to compare them when you have a different attack budget for bijection learning?
- **The presentation of results is poor as the figures are too small and contain too much data.** In general, I’d recommend thinking about the main insight you want the reader to take away from each figure and majorly simplifying it so that is the case.
    - Figure 3 - this is a little messy. Why do we need the tables on the right when the bar charts have all the details? I think it would be great to have a bar chart with the best-fixed size for each bijection type and compare directly against baselines for each model (using the full 320 behaviors since HarmBench-35 isn’t large enough). Then, a separate figure that ablates the fixed size. Then, it makes it easy to highlight the insights you’d like to convey in the caption and main text. What is the attack budget for these?
    - I think Table 1 contains the exciting results. I’d suggest leading with this before Figure 3. I’d also like to see comparable PAIR or TAP ASRs on the HarmBench 320 and AdvBench 50 set in the table. I think it is less important to show the fixed size and attack budget and just show the ASRs vs the baselines. Also, maybe fix the attack budget to the same as the budget for PAIR so it is more directly comparable. Also, how did you choose the current attack budgets? Was that the plateau point?
    - Figure 4 left - This is an awesome plot, and I think you should make a bigger deal out of it.
    - Figure 5 - I think this could be better as a line plot. Perhaps choose to plot either ‘digit’ or ‘letter’ rather than both to simplify. Share legend for all. Also, I think just having three lines: successful attack, refusal, and unhelpful non refusal would help simplify this. Potentially, just plotting unhelpful non-refusal for each model would get your point that smaller models can’t learn complexity bijections the best.
    - Figure 6 - share the legend so each plot is bigger. The main point you are trying to make is that the capability degrades as fixed size increases, but this message is easily lost with the amount of information you present. I’d recommend simplifying - just show it averaged over models and maybe drop some of the multi-digit results. You can put this plot in the appendix to refer to some nitty gritty details.
    - Figure 7 - I don’t think this is a scaling law. It is more of a Pareto frontier plot where you want to show the trade-off in two metrics. A scaling law would involve the amount of computing, data size, or size of the model. Did you experiment with different fits? A quadratic might not be as good as a higher-order polynomial. Why is a quadratic the right fit for this data?
    - General plotting advice: Please use error bars for all plots that take into account the sample size used (this can just be a standard error of a proportion that doesn’t involve rerunning many seeds). Use a colour-blind pallet, and make the font in figures a similar size as the font of paper.

**Questions:**

Most of my questions are in the weaknesses section, but here are some other questions/discussion points.

- What does “endless” mean? How is it different from any “adaptive” attack (like PAIR or random search suffixes) that will craft many potential jailbreaks until it finds one that works? I’m not sure emphasising this makes sense. Maybe you reframe it as an “Adaptive in context learning attack”?
- In the abstract, what does universal mean? Is it transferred across harmful requests or across models? Do you have a % of how the same bijection template transfers across requests and across models?
- Multi-turn conversation history - do you think it would be best to use the term few-shot learning here and then, in methodology, talk about how many shots you use (it looks like it is 10-shot?). Also, you call it “teaching examples” later. I would use few-shot since it is clear what it means.
- What defenses work / don’t work against your attack?
- Intro: I wouldn’t describe perplexity filtering, paraphrasing, etc as an alignment technique like RLHF. I’d describe it as an adversarial defense against jailbreaks.
- Maybe add the GPT-4o rewriter to correct typos in Figure 1, so it is clear how it works without needing to read the paper.

This paper has the potential for a much higher rating, but not in its current form. I would happily raise my score if the claims I mention in the weaknesses section are better explained and the results are significantly simplified in the figures and presented well. In addition, I’d like to see the comparison to adaptive black-box attack baselines like PAIR with a comparable attack budget.

---

> ### Author Response · Authors · 2024-11-22
>
> Thank you for your helpful and constructive feedback. We appreciate your suggestions and have significantly revised the paper to accommodate them, particularly the introduction (section 1), related work and contributions (1.1-1.3), and main ASR experiments (3.2-3.3).
>
> >Claims need to be better explained
>
> By and large, we agree with your suggestions in this section and have revised the introduction to be much more careful about making these claims, either by making them more specific or removing them from the paper. Below, we will try to answer the remaining relevant questions about our work.
>
> >Is your method for jailbreaking novel? Ciphers are not new, but perhaps your version of bijection learning in context is novel
>
> See our general response to reviewers for our portrayal of our novelty for more context. We also reframe this in our revised related works section (1.1) and contributions section (1.3).
>
> >What is the smallest LLM you tried the approach on? You say “new language model risk factors may emerge with scale” but also say the approach is “scale agnostic”.
>
> We agree that since our focus is frontier models, the smallest LLM we try is GPT-4o-mini and the approach is untested on even smaller models. The claim we wish to make is closer to the first one, and we’ve modified our wording accordingly. As an informal note, we've successfully used low-difficulty bijection attacks (namely, fixed size 24, or dispersion 2 in the current wording) on smaller models such as llama-3.1-8b-instruct and phi-3-instruct in the context of online redteaming arenas, although we have not benchmarked our attack on such models.
>
> >how many templates did you test, and how many of them worked?
>
> Our template went through at most a handful of iterations, and we did not try to optimize our template in particular. All the templates we tried worked, to varying extents.
>
> >Perhaps compare “egregiousness” with other jailbreaks and see if bijection learning induces more harmful ones. Otherwise, this claim isn’t interesting
>
> We do observe that bijection learning appears to induce more harmful responses than other jailbreaks on average. However, we find this claim hard to quantify, since “egregiousness” is subjective and variance among evaluators is high, so we remove it from the paper.
>
> >Our work is the first to produce attacks that are simultaneously black-box, automated, and scale agnostic” - I don’t think this is the case
>
> We agree and have changed this claim to “black-box, *universal*, and scale-adaptive.” See updated section 1.2 for more context.
>
> >“[whitebox methods] fail to transfer to the newest” - could you cite this? There is some GCG transfer
>
> We add comparison to GCG transfer in our main experiments; see updated Table 1.
>
> >it is unclear if you use an input+output classifier if they will catch your attacks or not. The classifier must be at the same capability level as the generation model
>
> We add guard model experiments in Section 5, with input and output classifiers protecting Claude 3.5 Sonnet. We find that if a less capable model is the guard model, it may fail to distinguish between harmful and benign prompts in bijection language. Surprisingly, even using Claude 3.5 Sonnet itself as the guard model only reduces ASR by 10 percentage points, so using input and output classifier models at the same capability level as the generation model may not be sufficient to prevent bijection attacks. See the updated Section 5 for more context.
>
> >Related work lacks comparing and contrasting with their work
>
> We’ve updated the related works section (1.1) to only include the most relevant works and compare to them in more detail.
>
> >Lack of threat model: Why are you working on black-box attacks on frontier models? Why is this more impactful to work on than white-box attacks? (I think it is)
>
> As model developers trend toward restricting access to internal components like tokenizers and logits, the black-box setting is a more realistic threat model. We note this in the updated related works section.
>
> More details in next comment

---

> ### Author Response · Authors · 2024-11-22
>
> >Fixed size (section 2.2)...I think it would be easier to define complexity, C, as the number of characters that map to something else
>
> We agree, and we call “dispersion” the number of characters that map to something else, since we also use other parameters like encoding length to adjust a bijection’s overall complexity
>
> >Report the false positive rate when you talk about it.
>
> We compare evaluation results with/without human filtering in Appendix C.
>
> >Also, do you check every single response in all your experiments?
>
> Yes.
>
> >Some more clarity on your method here (including a rubric your human evaluators follow)
>
> We add color on our criteria for human evaluators at the end of Section 3.1.
>
> >Add how you filter HarmBench, e.g. do you just use standard direct requests and remove copyright/context ones?
>
> We use the entire test set, including copyright and contextual requests.
>
> >There should be more baselines, e.g. vs PAIR and TAP. You could evaluate these with the BoN methodology too.
>
> We agree. We evaluate PAIR with the same attack budget as bijection learning and TAP with a slightly higher attack budget. See updated Table 1.
>
> >I expect PAIR to get similar ASRs on GPT4o-mini
>
> Bijection learning gets an ASR of 64% vs PAIR’s 23%. The PAIR paper doesn’t report results on GPT4o, but TAP does. However, TAP’s results are “as of May 2024,” implying they tested an early release, while we ran experiments on the mature 2024-08-06 release which has more safety tuning. This may explain why the new ASRs are much lower than previously reported (plus we run TAP with lower than the original attack budget for budget parity with bijection learning). We run TAP’s original code, so there is no discrepancy in implementation.
>
> >Please explain your implementation for ascii, caeser-cipher, morse code and self-cipher…Is it fair to compare them when you have a different attack budget for bijection learning
>
> We’ve replaced these baselines with an ensemble baseline of 11 encoding-based attacks, described in Section 2.2, effectively simulating an attack budget of 11 for previous attacks to make the comparison more fair. See general response to reviewers for more context.
>
> >The presentation of results is poor as the figures are too small and contain too much data
>
> We generally agree with your comments and have adjusted our figures accordingly.
>
> >I don’t think this is a scaling law. It is more of a Pareto frontier plot where you want to show the trade-off in two metrics. A scaling law would involve the amount of computing, data size, or size of the model.
>
> We agree that it is more accurately a “Pareto frontier” and have adjusted our discussion accordingly. However, we believe that the constant “scaling law” it indicates is that bijection learning is most effective when the target model has been degraded to a MMLU accuracy of 60%, regardless of model scale.
>
> >Did you experiment with different fits? A quadratic might not be as good as a higher-order polynomial. Why is a quadratic the right fit for this data?
>
> A quadratic fit is the lowest-parameter model that can capture the trend in which ASR increases with MMLU accuracy and then decreases. We do try higher-order fits but they do not help the fit much.
>
> >What does “endless” mean? How is it different from any “adaptive” attack (like PAIR or random search suffixes) that will craft many potential jailbreaks until it finds one that works? I’m not sure emphasising this makes sense. Maybe you reframe it as an “Adaptive in context learning attack”?
>
> We use the word “endless” to distinguish our attack from previous encoding-based attacks in particular. Our sampling method for bijections allows us to perform best-of-$n$ sampling from an endless pool of encodings, while previous encoding-based attacks only create a single prompt for each intent.
>
> >In the abstract, what does universal mean? Is it transferred across harmful requests or across models? Do you have a % of how the same bijection template transfers across requests and across models?
>
> “Universal” refers to transferability across harmful requests. Bijection templates likely do not transfer well across models, since each model has its own ideal bijection complexity parameters.
>
> >do you think it would be best to use the term few-shot learning here
>
> We generally agree, but should be a bit careful because the teaching shots in bijection learning are not really instances of the same task we are asking the model to perform, which is to respond to a bijection language query in bijection language, not translate from English to bijection language. Only the MMLU practice examples are “true” teaching shots.
>
> >What defenses work / don’t work against your attack?
>
> See discussion of guard model experiments above. Guard models do not seem particularly effective against our attack (unless we allow guard models to refuse all enciphered text).
>
> Overall, we are grateful for the quantity and quality of suggestions, and look forward to hearing back from you.

---

> ### Author Response · Authors · 2024-11-25
>
> Dear reviewer, just wanted to send a gentle reminder that we've posted our rebuttal. We greatly appreciate your extensive notes in your original review and have made significant revisions to our exposition and visual presentation; added more comprehensive baselines; and added new experimentation around guardrails as a possible defense for our attack. We'd like to know if our response resolves your concerns, and we hope you'll consider strengthening your support for our work.

---

> > ### Comment · Reviewer_sNn1 · 2024-11-25
> >
> > Thanks very much for addressing my feedback! I think the paper is now significantly better than before. The presentation is improved, and the extra baseline experiments strengthen the impact of your method. I will be raising my score as a result.

---

> > > ### Author Response · Authors · 2024-12-04
> > >
> > > We would like to thank the reviewer for engaging deeply with the paper, providing meticulous feedback, and ultimately extending their support for our submission. We’ve found your notes invaluable for iterating from our original manuscript into a significantly more polished final draft, and we reiterate our deep appreciation for the reviewer’s unique effort.

---

### Official Review · Reviewer_53mm · 2024-11-02

**Soundness:** 3
**Presentation:** 3
**Contribution:** 2
**Rating:** 8
**Confidence:** 3

**Summary:**

This paper proposes a new method for jailbreaking LLMs by means of communicating in an encrypted format.
The LLM is first given a simple encryption algorithm for encrypting and decrypting strings.
Then, the user and LLM communicate for several rounds in this encrypted format.
After some time, the user will ask a malicious query (e.g., "Design a weapon") in this encrypted format; this paper finds that the LLM response, when decrypted, is often a jailbroken one.
This style of attack, dubbed a "bijection attack", is more successful against stronger LLMs.

**Strengths:**

The authors present an interesting attack that demonstrates the surprising ability to scale with LLM power: stronger LLMs appear more susceptible to this attack. Moreover, the proposed attack achieves key desiderata of jailbreaks: against a black-box target, universal/automatable, and scalable. These qualities make "bijection attacks" a valuable benchmark for LLM developers to consider when evaluating safety.

**Weaknesses:**

In my opinion, this paper does not have clear weaknesses. However, given the state of jailbreaking research, I do not think that this style of attack paper is scientifically or technically exciting. To change my opinion, I would like to see some deeper technical insights + experiments, possibly with the authors' proposed defense strategies in Section 5 --- but this may be unreasonably ambitious in the rebuttal time frame. While my impression leans on the negative side, I am okay with accepting this work if the other reviewers do not have strong objections.

**Questions:**

It would be good to see additional discussion on how LLM developers might defend against this style of attack.

---

> ### Author Response · Authors · 2024-11-22
>
> Thank you for your constructive feedback.
>
> >this paper does not have clear weaknesses. However, given the state of jailbreaking research, I do not think that this style of attack paper is scientifically or technically exciting
>
> While we’re sorry that you feel this way, this seems like more of a critique than an entire field than our specific paper. Previously, similar-style papers discussing black-box jailbreak prompts have commonly appeared in top conferences, including [1] (NeurIPS ‘23), [2] (NeurIPS ‘24), and [3] (ICLR ‘24), some of which include minimal discussion of defense mechanisms.
> [1] https://openreview.net/forum?id=jA235JGM09
> [2] https://openreview.net/forum?id=cw5mgd71jW
> [3] https://openreview.net/forum?id=MbfAK4s61A
>
> >To change my opinion, I would like to see some deeper technical insights + experiments, possibly with the authors' proposed defense strategies in Section 5 --- but this may be unreasonably ambitious in the rebuttal time frame
>
> We appreciate the suggestion and have added additional experiments to the paper (see the new Section 5) that showcase the defense strategy we previously proposed but did not test. In particular, we consider defending a target model (Claude 3.5 Sonnet) by using some guard model (we test GPT-4o-mini, Claude 3 Haiku, and Claude 3.5 Sonnet) to filter the inputs and outputs of the target model. The guard model has access to the same information as the target model (the entire conversation history) when acting as an input filter, but is prompted to reject harmful prompts. As an output filter, the guard model has access to the entire conversation history *in addition* to the model response. We use HarmBench and a dataset of 100 benign prompts (harmless questions like "how do I cook spaghetti?") with our bijection learning prompt.
>
> Using less capable models should in theory not mitigate the attack much, since the target model can understand bijections that the guard model cannot, and in the case the guard model cannot do better than guessing. Indeed, GPT-4o-mini stops 9% of harmful queries but 12% of benign queries, while Claude 3 Haiku rejects nearly all queries, harmful or benign. Surprisingly, using Claude 3.5 Sonnet as its own guard model only decreases bijection learning ASR by 10 percentage points when using a bijection learning attack budget of 6, even though it is able to understand the same bijections. Adding the same model as a guard also triples inference costs (since we need to call Claude 3.5 Sonnet thrice instead of once for each query).
>
> See the revised Section 5 for more. While using guard models is just one defense strategy, it is one of the most applicable defenses in the real world, since it is a "black-box" defense. Other defenses (such as adversarial training; representation engineering; and fine-tuning against unsafe data in various ways) are impossible for us to implement on frontier models since they often require access to model internals. However, LLM developers themselves may be effective at implement these strategies..
>
> Please let us know if you have any additional questions or feedback. We hope that you would consider increasing your score.

---

> > ### Comment · Reviewer_53mm · 2024-11-22
> > **Response**
> >
> > Thanks for the additional experiments and discussion. These changes have improved the paper according to the criterion that I listed, and I have likewise increased my score.
> >
> > However, I stand by my opinion that coming up with yet another prompt-based jailbreak is not a useful contribution in 2024 because there are too many existing attacks that remain poorly understood. To end on a more positive note, I believe that the authors are very well-positioned to investigate more meaningful directions, like better understanding jailbreak failure mechanisms and defense techniques.

---

> > > ### Author Response · Authors · 2024-11-25
> > >
> > > Thank you for your continued engagement and for acknowledging that our additional experiments have improved your opinion of the paper.
> > >
> > > We do think the portrayal of our work as “yet another prompt-based jailbreak” is a bit unfair. Unlike other work on prompt-based jailbreaks, we perform extensive analysis to understand our proposed attack and why models are vulnerable to similar attacks, including but not limited to our new experiments on guard models, adjusting quantitative attack parameters, classifying failure modes, and analyzing capability degradation at different model scales. Thus, we do think that our work, in particular our computational overload argument for the attack efficacy of bijection learning, provides more explanatory power and gets closer to exposing fundamental sources of model vulnerability than previous work on prompt-based jailbreaks.
> > >
> > > Re: adding more defenses, we do agree that further analysis of defenses would be useful but are limited by the time constraint of the rebuttal period. Given more time, we will continue to add a greater variety of defenses and a greater depth of analysis.
> > >
> > > Thanks again for devoting your time to the improvement of our work.

---

> > > > ### Comment · Reviewer_53mm · 2024-11-25
> > > >
> > > > Alright, you've sold me. 6 -> 8.

---

> > > > > ### Author Response · Authors · 2024-12-04
> > > > >
> > > > > We thank the reviewer for their engagement in our rebuttal and for their thoughtful comments regarding model defenses and future directions. We are grateful for the increase in score and for the reviewer’s advocacy of our paper.
> > > > >
> > > > > Overall, we agree with the reviewer that investigating underlying jailbreaking mechanisms and building more robust model defenses are pertinent and high-impact directions to pursue. This “blue-teaming” research does not occur in a vacuum, but rather, blue-teaming and red-teaming research synergize: investigating attacks gives valuable intuitions and insights about defenses, and vice versa. We believe our work contributes meaningfully to this combined approach.

---

### Official Review · Reviewer_WL23 · 2024-11-03

**Soundness:** 2
**Presentation:** 3
**Contribution:** 2
**Rating:** 3
**Confidence:** 4

**Summary:**

This paper proposes a general framework for jailbreaking attacks with bijection encoding ciphers. Experiments show that the attack is effective on a wide range of frontier language models. They also find out that the bijection cipher attack is more effective on larger and stronger language models.

**Strengths:**

1. ASR of the proposed method is high on many frontier LLMs.
2. The authors did comprehensive experiments to verify the effectiveness of their method. The results reported in Section 3.3 is interesting.

**Weaknesses:**

1. The novelty of this work is questionable given many existing cipher based jailbreaking attacks. It seems the only difference between this paper and existing works is that this paper proposes to use a system message to customize general cipher encodings. I'm not confident about whether the contributions are enough for ICLR.
2. Comparisons between many other cipher-based jailbreaking attacks are missing, including but not limited to:
[1] When “Competency” in Reasoning Opens the Door to Vulnerability: Jailbreaking LLMs via Novel Complex Ciphers
[2] Jailbreaking Large Language Models Against Moderation Guardrails via Cipher Characters

**Questions:**

Please see the weakness part above. Could the authors explain more on how their proposed methods differ from the existing ones?

---

> ### Author Response · Authors · 2024-11-22
>
> Thank you for your constructive feedback.
>
> >The novelty of this work is questionable given many existing cipher based jailbreaking attacks. Could the authors explain more on how their proposed methods differ from the existing ones?
>
> See general comment to all reviewers for our portrayal of the novelty and significance of our work, and how it goes further than previous cipher-based jailbreaking attacks. We note that:
>
> *Previous encoding-based attacks encode a harmful prompt in some particular encoding, like Base64, ASCII, or ROT13, that the model already knows about through training. On the other hand, our method involves teaching models new encodings through in-context learning. Compared to previous work:*
>
> *1. Our sampling method is a powerful fuzzing technique that allows us to perform best-of-$n$ sampling from an essentially endless pool of possible encodings, which doubles our ASR compared to using only 1 random encoding. Best-of-$n$ sampling is not possible using previous techniques – an attacker can keep trying more and more well-known or hand-crafted ciphers, languages, etc. proposed in other papers, but will eventually run out of options.*
>
> *2. We use quantitative parameters to scale the encoding difficulty to best match the model’s capability level. In particular, our dispersion parameter d, 0<=d<=26, controls how many different letters are mapped to an encoded sequence, where d=0 is plain text and d=26 permutes all letters. Previous attacks are rather blunt: either the model knows the selected encoding or it doesn’t. However, for bijection learning, if a certain model cannot learn to permute all the characters, we can tone down the dispersion to map only a few letters to something else. Hence, our method is uniquely scale adaptive in a way that other methods are not.*
>
> Notably, we believe that we are the first work among any black-box jailbreaking work, not just work on encoding jailbreaks, to adjust our attack prompt smoothly using quantitative parameters to adjust to varying model scales. This allows us to analyze how attack efficacy and failure modes smoothly transition with bijection complexity. We derive the surprising relationship that bijection learning is most efficacious when model capabilities are degraded to about 60% performance on MMLU, regardless of model size.
>
> We’ve also updated our related work and contributions sections (1.1-1.3) to reflect this, and encourage you to revisit these revised sections.
>
> >Comparisons between many other cipher-based jailbreaking attacks are missing
>
> We agree that we can do more to compare to previous cipher-based jailbreaks, and we do so in both our updated introduction (1.1-1.3) and our experiments (see section 3.2 and Table 1). We revise the paper to include the encodings (keyboard, upside down, word reversal, and grid) considered in [1] to our ensemble baseline of 11 encoding-based jailbreaks. We also include in this ensemble a variety of other encoding-based attacks found in previous work, including https://arxiv.org/abs/2307.02483, https://arxiv.org/abs/2402.11753, https://arxiv.org/abs/2308.06463. We mark this ensemble baseline successful for an attack intent if at least 1 of 11 encodings caused a successful jailbreak. **Bijection learning outperforms this ensemble baseline by at least 30 percentage points on all models.**
>
> Even though the title of [2] includes the word “cipher,” the paper does not present an encoding-based jailbreak. Instead, the paper presents a white-box optimization attack that involves inserting a substring between words, and is only tangentially related to our work.
>
> We would appreciate any additional questions or feedback that you have, and we hope that you would consider increasing your score.

---

> ### Author Response · Authors · 2024-11-25
>
> Dear reviewer, with the review period coming to a close, we wanted to send a gentle reminder that we've posted our rebuttal alongside a significantly revised paper. We'd like to highlight that other reviewers have significantly increased their scores, which we believe is a sign that our current revision makes meaningful improvements upon the original version of the paper.
>
> We believe that we’ve clarified in our rebuttal how our methods and analysis extend significantly beyond previous work, and we've added comparison to several previous cipher-based attacks, including the work you listed, as part of an ensemble baseline. We've also added many additional baselines beyond your comment, namely GCG, PAIR, and TAP, and new experiments showing that our attack is robust to guard model defenses.
>
> We are open to further discussion, and if our revisions and clarifications have addressed your main concerns, then we hope you would reconsider your original score. Thank you for taking the time to review our work.

---

> > ### Comment · Reviewer_WL23 · 2024-12-01
> >
> > Thanks to the authors for additional discussions.
> >
> > I agree with the authors that the proposed method reaches a much higher ASR compared to existing methods and the experiments are comprehensive. However, I still feel the contribution of this work is limited as it heavily builds upon the existing encoding-based works. It basically shows the encoding-based attacks can be stronger when more different encodings are available.  In all, I'm still not convinced about the significance of the contributions of this work.
> >
> > Some minor points:
> > > Notably, we believe that we are the first work among any black-box jailbreaking work, not just work on encoding jailbreaks, to adjust our attack prompt smoothly using quantitative parameters to adjust to varying model scales.
> >
> > This statement might be overclaimed. Some other attacks such as pre-filled attacks can also control the strength of attack with the length of pre-filled length.

---

> ### Author Response · Authors · 2024-12-02
>
> Thanks for your additional discussion and your recognition that our work produces empirical improvements upon existing encoding-based attacks and more comprehensive experiments and analysis.
>
> >It basically shows the encoding-based attacks can be stronger when more different encodings are available.
>
> We agree that the idea that using an endless quantity of different encodings can make the attack stronger is one aspect of our work, but we’d like to emphasize that we think this framing is a bit reductive and our contributions extend greatly beyond making this point.
>
> One main takeaway is that encoding-based attacks are scale-adaptive, e.g. for every model there exists some ideal encoding complexity level for jailbreaking the model. Even if we restrict our attack to sample only 10 different encodings, a similar number to the ensemble baseline (which uses 11 encodings), we achieve significantly better ASRs on all models (by at least 20 percentage points); see Figure 3. Sampling just one encoding beats the ensemble baseline on Claude models. Thus, our attack owes at least as much of its success to its scale-adaptivity as its use of different encodings.
>
> Additionally, we provide evidence for a previously unexplored jailbreak mechanism, which is that encoding translation takes up the model’s capabilities and thus degrades the model’s ability to perform safety interventions. This effect means that safety-capability parity between a guard model and a target model may not suffice to make the target model safe; see Section 5. Also see our extended discussion with Reviewer CzKw, where we give updated analyses regarding our scaling laws and jailbreak defenses and discuss our evidence for capability degradation as a jailbreak mechanism. (We are no longer allowed to edit the paper to include our new experiments and figures, but we will add these in the finalized paper.)
>
> >This statement might be overclaimed. Some other attacks such as pre-filled attacks can also control the strength of attack with the length of pre-filled length.
>
> We do not presently make that statement in the paper, so it is indeed a minor detail, but we currently believe that it is true.
>
> For your pre-filling example, while it is possible to vary the prefill length, we believe that an argument about scaling is not made in prior work (e.g. https://arxiv.org/html/2404.02151v1, which only uses prefill lengths up to 25 tokens). To our best knowledge, it is not argued that varying prefill length helps to adapt the attack to different model scales, e.g. that longer prefills are more effective on stronger models while shorter prefills are more effective on weaker models. We don't think it's clear why such a dynamic would hold.
>
> If you have any examples in mind for similar scaling claims made in other work, we would sincerely appreciate additional detail or citations.
>
> Thanks again for your continued discussion, and we would appreciate it if you would let us know if you have any other questions or comments.

---

> ### Comment · Reviewer_WL23 · 2024-12-02
>
> Thanks for the additional discussions. I've read the "extended discussion with Reviewer CzKw, where we give updated analyses regarding our scaling laws and jailbreak defenses and discuss our evidence for capability degradation as a jailbreak mechanism".
>
> However, I'm still not convinced that the additional discussions are insightful enough for the jailbreaking attack/defense community. There are too many similar works improving encoding-based, reasoning-based, replacement-based jailbreaking attacks that mix the jailbreaking attack with a more difficult task and increase the attack success rate. I randomly list some references below but they are apparently not exhausted.
>
> Therefore, although I appreciate the comprehensive experiments done by the authors, I expect newer insights provided by jailbreaking papers if the attack method itself is not new enough.
>
> [1] Cognitive Overload: Jailbreaking Large Language Models with Overloaded Logical Thinking
>
> [2] DrAttack: Prompt Decomposition and Reconstruction Makes Powerful LLM Jailbreakers

---

> ### Author Response · Authors · 2024-12-03
>
> Thanks for your continued engagement. We continue to stand by the novelty and significance of our contributions.
>
> [1] is motivated by a vague concept of “cognitive overload” and proposes jailbreaks with this idea in mind, but they do not provide any evidence that this is the actual mechanism that facilitates their jailbreaks. Their most relevant jailbreak is switching between English and another language in the prompt. But their Figure 11 shows that on the majority of models tested, this approach achieves similar or lower ASR than conversing entirely in that other language, so its success is clearly explained by mismatched generalization. Their jailbreaks are motivated by an overly anthropomorphic theory of LLM "cognition" rather than a benchmarked analysis of model capabilities.
>
> Unfortunately, we fail to see how [2], which inserts filler tokens between words in an attack prompt, represents an attempt to overwhelm the model’s computational capacity; we do not believe this argument is made in the paper. We'd appreciate any additional clarification here.
>
> In general, it may be the case that many people have had a similar idea of how a possible jailbreak predicated on computational overload may work, so much so that it may already seem like a foregone conclusion that one exists, but to our best knowledge, prior work does not produce reasonable empirical evidence that computational overload serves as the mechanism for some jailbreak. If we may proffer, we believe some of the most significant work in science occurs when an effect that has long been speculated to exist is actually shown to exist. In our case, knowing how the computational overload mechanism materializes can allow model developers to better study it and take it more seriously as a potential threat to model safety.
>
> While we appreciate your effort to “randomly list some references,” we welcome you to raise any particular high-quality work that you believe has already covered the same ground as our work, because we believe it would lead to a more productive discussion.
>
> We’re glad you agree that our experiments are comprehensive. We hope you will be convinced that our analysis is sufficiently insightful for the community (as some other reviewers seem to think).

---

> > ### Comment · Reviewer_WL23 · 2024-12-03
> >
> > Thanks for the update.
> >
> > But as I have mentioned before, the two references are just some examples for existing works on "too many similar works improving encoding-based, reasoning-based, replacement-based jailbreaking attacks that mix the jailbreaking attack with a more difficult task and increase the attack success rate". I list more references below on increasing the jailbreaking success rate by embedding it within a difficult task:
> >
> > [1] A Wolf in Sheep’s Clothing: Generalized Nested Jailbreak Prompts can
> > Fool Large Language Models Easily
> > [2] CodeChameleon: Personalized Encryption Framework for
> > Jailbreaking Large Language Models
> >
> > Besides, I have additional questions over the soundness of additional discussions about Capability degradation and Mismatched generalization.
> > > Thus, peak ASR should coincide with MMLU scores that are around the level of the pre-trained model.
> >
> > However, why does an instruct-tuned model with a certain level bijection learning match the status of a pretrained model?

---

> ### Comment · Reviewer_WL23 · 2024-12-03
>
> Some additional comments about the extended discussion:
>
> > Even a bijection with dispersion of 2 or 3 diverges significantly from English plaintext and constitutes a writing task not present in most pre-training distributions. If mismatched generalization were to explain bijection learning efficacy, then low-dispersion bijection attacks, i.e. easy bijections that produce little-to-no computational overload as evidenced by MMLU evaluations, should obtain significant ASRs.
>
> This is not a strong statement. How do the author define "significant ASRs"? As current models are mostly finetuned with slightly noisy data, it's expected that they can resist slight perturbations.
>
> In all, the extended discussion looks not convincing to me as well. However, it could be from my failure of correctly understanding the discussion.

---

> ### Author Response · Authors · 2024-12-03
>
> Thank you for the quick response and for continuing to engage in discussion with us.
>
> >I list more references below on increasing the jailbreaking success rate by embedding it within a difficult task.
>
> To be clear, only one of the four works you’ve listed is about embedding an attack inside a more difficult task, namely [2] from this latter comment. But again, [2] does not give the same breadth of experiments or depth of analysis, and they do not attempt to show that their attack succeeds due to degrading the model’s capabilities. Instead, they observe that “LLMs excel at deciphering the encrypted queries,” unlike our approach in which we increase the difficulty until the LLM is on the brink of no longer being able to reason, and so their approach is more similar to obfuscation or persuasion-based approaches that trick the model into misclassifying the prompt rather than degrading its ability to classify unsafe prompts in general.
>
> >Thus, peak ASR should coincide with MMLU scores that are around the level of the pre-trained model.
>
> >However, why does an instruct-tuned model with a certain level bijection learning match the status of a pretrained model?
>
> Mismatched generalization is defined as the following:
>
> “Pretraining is done on a larger and more diverse dataset than safety training, and thus the model has many capabilities not covered by safety training. This mismatch can be exploited for jailbreaks by constructing prompts on which pretraining and instruction following generalize, but the model’s safety training does not. For such prompts, the model responds, but without safety considerations.” (https://arxiv.org/pdf/2307.02483)
>
> If mismatched generalization were the primary mechanism for the bijection learning jailbreak, then we should see the attack reach peak ASR when we pass a prompt on which fine-tuned safety capabilities do not generalize, but pretrained capabilities do generalize. In other words, mismatched generalization is the idea that there exists a prompt that causes the model to respond as if it were in its pre-trained state (thus, its capabilities should be at least that of the pre-trained model).
>
> While we would see a slight decline in MMLU performance from the model behaving in this state (since the model is better at MMLU after fine-tuning), degrading the model’s capabilities past the level of the pre-trained model should not benefit the attack. However, empirically, we see that further degradation **does** benefit the attack, showing that our peak ASR is related to capability degradation, not mismatched generalization.
>
> >How do the author define "significant ASRs"? As current models are mostly finetuned with slightly noisy data, it's expected that they can resist slight perturbations.
>
> Thanks for the question, we think our wording in the original response was a bit unclear. The plot of Gemini 1.5 models illustrates what we expect our Pareto frontier to look like in general, if the underlying mechanism is mismatched generalization: bringing the model OOD with respect to its safety-training data should immediately achieve near-peak ASRs.
>
> On the other hand, letter bijections on GPT4o with dispersions 8-10 and letter bijections Claude 3.5 Sonnet with dispersions 16-26 achieve less than half of the peak ASR on their respective models because they do not sufficiently degrade model capabilities (still >90% MMLU). These settings cannot be construed to consist of slightly noisy data: they are essentially unrecognizable as plain English. For example, here’s a sentence with dispersion 10: “shke! gw hse ekkwk-baseo sql indecgiwn, ywh mhsg fksg ckeage a cwnneciwn gw ywhk pwsgukesql oagabase.”
>
> **More generally**, we argue that the *sufficiency* claim (I) alone is enough to show that capability degradation is an important factor in the efficacy of bijection learning. Mismatched generalization is about turning off the model’s tendency to classify and reject harmful prompts. But we’ve shown through our guard model experiment in Section 5 and our new experiment on safety classification scaling with MMLU accuracy that even if the behavior to classify harmful prompts still exists, the model is ineffective at classifying harmful prompts when conversing in bijection languages at the right complexity level. Thus, our results show that *even if safety guardrails completely generalize across all possible prompts, which may be the case for certain models like Claude or GPT for which the developers tried really hard to robustify their safety training, these guardrails can be ineffective for attacks like bijection learning due to degradation of capabilities.*

---

> > ### Comment · Reviewer_WL23 · 2024-12-03
> >
> > I still feel the additional explanations are not convincing to me with lots of unclear statements. For example:
> >
> > > While we would see a slight decline in MMLU performance from the model behaving in this state (since the model is better at MMLU after fine-tuning), degrading the model’s capabilities past the level of the pre-trained model should not benefit the attack.
> >
> > Why does this support the claim that “our peak ASR is related to capability degradation, not mismatched generalization.”? When the difficulty of bijection learning increases and MMLU performance decreases past this point, it's reasonable that ASR would continue to increase as the difference between encoded texts and already-seen distribution gets larger.
> >
> > In all, at the current point, I don't feel the evidence and discussions provided by the authors are convincing enough to me with many unclear statements from my understanding. Therefore, I left the evaluation of these additional discussions to other reviewers and I'll lower my confidence score.

---

> ### Author Response · Authors · 2024-12-03
>
> Thank you for the continued discussion.
>
> According to a mismatched generalization hypothesis of our jailbreak, fine-tuning on a distribution of English plaintext simultaneously causes both
>
> (a) A bump in MMLU accuracy (around 5-10 percentage points)
>
> (b) Refusal to answer unsafe prompts
>
> Low dispersion settings (e.g. dispersion 6-8 for Claude 3 Sonnet and Gemini) cause the model’s MMLU to revert to approximately pre-trained level. (Note, these settings are easy enough that they do not cause broader capability degradation of the pre-trained model capabilities, as indicated by a lack of increase in incoherent answers: https://anonymous.4open.science/r/bijection-rebuttal-E525/more-failure-modes.png) Per the mismatched generalization hypothesis, this reversion to pre-trained level reflects that the data is sufficiently out-of-distribution such that the model’s fine-tuned capabilities do not generalize. Thus, we should observe that peak ASR is achieved around this point. However, we only achieve less than half of peak ASR on Claude and GPT models, indicating that it is further degradation of the pre-trained capabilities that is responsible for most of the efficacy of our jailbreak.
>
> Furthermore, we disagree with the general idea that lack of generalization scales smoothly with the difference between encoded and in-distribution text. If that were true, on Gemini models, for which we believe jailbreak efficacy is adequately explained by mismatched generalization, then ASR should increase with dispersion as the text becomes more out-of-distribution. Instead, we see an immediate jump in attack success between in-distribution and out-of-distribution text at the lowest dispersion setting of 2, and attack success doesn’t increase further as dispersion increases, even though the text may be more out-of-distribution under some metric.
>
> We hope you feel your questions have been clarified. At the very least, we acknowledge that our responses have decreased your confidence in your criticism.

---

### Official Review · Reviewer_CzKw · 2024-11-03

**Soundness:** 3
**Presentation:** 3
**Contribution:** 2
**Rating:** 6
**Confidence:** 4

**Summary:**

The paper discusses an approach to jailbreaking via bijection learning. Specifically, they generate a random transformation over characters that shifts the input prompt out of the safety fine-tuning distribution. They find that models prompted to answer inputs in bijection format are more likely to output harmful content compared to standard model inference.

**Strengths:**

1. The algorithm is clear, simple, and admits random sampling for endless bijections.
2. The analysis is comprehensive. I really appreciated the scaling analyses for 1) the n in best-of-n and 2) the ASR vs model capabilities frontier showing that more capable models may be more susceptible.

**Weaknesses:**

1. The main ideas presented in this paper have been identified in prior works under mismatched generalization [1] and lack of robustness to out-of-fine-tuning distribution prompts such as low-resource languages [2,3]. [1] also makes the observation that transformation-based jailbreaks benefit from increasing model scale. As such, this paper extends these ideas rather than introduces them.
2. I believe the comparison in Figure 3 to the baselines is not apples-to-apples since bijection learning performs a best-of-6 while the baselines are effectively best-of-1. I think a more fair comparison is either doing best-of-1, or having a combined baseline that's an ensemble of six reasonable baselines.

[1] https://arxiv.org/abs/2307.02483
[2] https://arxiv.org/abs/2310.02446
[3] https://arxiv.org/abs/2309.10105

**Questions:**

1. "We search for universal attack mapping candidates from our earlier evaluations by selecting attacks which produce a strong response for a particularly malicious intent. For a selected mapping, we generate the fixed bijection learning prompt and evaluate it as a universal attack on HarmBench" (quote starting from line 342). I was confused by this phrasing, is the scaling plot in Figure 4 for best-of-n at n=1 done for a single random sample of a language, or is it done for the single best language subselected from all the prior experiments?

---

> ### Author Response · Authors · 2024-11-22
>
> Thank you for your insightful and positive response.
>
> >The main ideas presented in this paper have been identified in prior works under mismatched generalization
>
> We agree that one reason that has been proposed for the efficacy of encoding-based attacks is that safety training does not generalize to out-of-distribution settings like low-resource languages. In addition, our work exposes a new explanation for the efficacy of our attack (which may, to a lesser extent, also be a factor in the efficacy of previous encoding attacks): translating to/from bijection language is a difficult task that the model must perform concurrently with giving its response, so conversing in bijection language makes the model less capable at other tasks (see Figure 6), including the task of classifying and rejecting harmful responses. We show this through our Figure 7, which indicates that even though the models studied have vastly different abilities for learning bijections, all of the models share the commonality that bijection learning is the most effective when the complexity is high enough to lower the MMLU capabilities of the target model to about 60%. We believe this finding illustrates that jailbreaks can succeed by “overloading the model’s computational capacity,” which is a different idea from mismatched generalization.
>
> >I believe the comparison in Figure 3 to the baselines is not apples-to-apples since bijection learning performs a best-of-6 while the baselines are effectively best-of-1.
>
> A key advantage of our approach over previous encoding-based attacks is that it is an effective “fuzzing” technique that enables us to generate many variants of our attack prompt, which enables best-of-n sampling that is naively not possible using other encoding-based attacks. However, we agree that it would make our comparison more compelling if we artificially gave previous work this degree of freedom, and we like your suggestion about simulating best-of-n by using an ensemble of baselines.
>
> We noted the following in our response to all reviewers:
>
> *We ensemble 11 of the most popular and best performant encoding baselines (8 more than before) and consider this ensemble attack successful for an attack intent if any one attack succeeds, similar to best-of-11 sampling. **Bijection learning still outperforms by over 30 percentage points on all models** (see Table 1). In fact, best-of-1 bijection learning still outperforms this ensemble on Claude models, and best-of-5 bijection learning outperforms on all models (see Figure 3).*
>
> >is the scaling plot in Figure 4 for best-of-n at n=1 done for a single random sample of a language, or is it done for the single best language subselected from all the prior experiments?
>
> The scaling plot in Figure 4 (now Figure 3) uses a single random sample of a language, not the single best language. We’ve made this plot standalone for the sake of clarity, and moved the table (which showed the single best language) to the appendix.
>
> Please let us know if you have any additional questions or suggestions for the paper, and we hope you will strengthen your support for our submission.

---

> ### Author Response · Authors · 2024-11-25
>
> Dear reviewer, with the discussion period coming to a close, we wanted to send a gentle reminder that we've posted our rebuttal alongside a significantly revised paper.
>
> We’ve adjusted our writing to highlight how our work goes beyond the previous literature, and we believe that our evidence of a “computational overload” argument for the efficacy of our jailbreak is substantially different from the mismatched generalization effect observed in previous jailbreaking work. Additionally, we’ve adjusted our baselines to contain an ensemble approach of a large number of ciphers to make the comparison more fair, as you suggested; and we’ve added many additional baselines beyond your comment, namely GCG, PAIR, and TAP.
>
> If our changes have addressed your concerns, we hope you would strengthen your support for our submission, as have several other reviewers. We are also open to further discussion.
>
> We look forward to your response. Thank you for taking the time to review our work!

---

> ### Comment · Reviewer_CzKw · 2024-11-25
>
> Thank you for the improvements, I believe they make the paper more rigorous. However, I do not follow the computational overload argument (even after reading the new sections in the paper and Figure 6/7). Specifically, I do not understand
>
> 1. Why a jailbreaking defense is seen as taking some level of computational capability at inference time
> 2. If all the peaks of the scaling curves are nearr 60, why would this indicate that bijection learning is taking some fixed computational capability
> 3. Are all the peaks of the scaling curves actually 60? (it looks like a 10% variation, and that too from the fitted quadratics which aren't a really good fit of the original points in the first place)
>
> I believe it could be interesting to show this hypothesis and that it's not mismatched generalization (if true). Unfortunately, it does not feel substantiated in the current version of the paper and my rating remains the same to reflect the empirical successes of the attacks over prior work.

---

> ### Author Response · Authors · 2024-12-02
> **Reply to reviewer (1/2)**
>
> Thanks for your continued engagement with our work. We will try to clarify the points of confusion.
>
> >A. Why a jailbreaking defense is seen as taking some level of computational capability at inference time
>
> Stronger models are better at performing classification of unsafe prompts. We verify this empirical fact through an additional experiment, which shows that safety classification accuracy scales with MMLU capabilities. We plot 5-shot MMLU accuracy against classification accuracy for various models on a random sample from the OR-bench dataset consisting of 2,000 safe prompts and 500 unsafe prompts: https://anonymous.4open.science/r/bijection-rebuttal-E525/classification-scaling.png
>
> It appears that models’ ability to separate safe from unsafe prompts is linked to their general reasoning capabilities. We are no longer allowed to edit the paper, but we will include this plot in an appendix section since it further illustrates this point.
>
> >B. If all the peaks of the scaling curves are near 60, why would this indicate that bijection learning is taking some fixed computational capability
>
> We do not claim that bijection learning takes some fixed computational capability. Instead, our argument is that *classification of unsafe prompts* requires some amount of computational capability. Specifically, we believe the ability to robustly classify unsafe prompts begins to emerge in models when their MMLU capabilities are above 55-70%. When their capabilities are degraded below this point, models become ineffective at rejecting unsafe prompts. However, their ability to provide relevant and helpful answers also decreases.
>
> With bijection learning, we adjust the complexity of the encoding until the model’s *remaining* capabilities are about enough to score 55-70% on MMLU, which is around the point where they are unable to robustly perform safety classification but are still able to give coherent answers. Stronger models have more computational capacity in general, so we use more complex encodings to take up more of their capacity until their capabilities are degraded to this point.
>
> >C. Are all the peaks of the scaling curves actually 60? (it looks like a 10% variation, and that too from the fitted quadratics which aren't a really good fit of the original points in the first place)
>
> We agree that our analysis could benefit from additional precision. In an updated analysis, we add:
>
> (a) additional data points for each model
>
> (b) more models (Claude 3 Sonnet, Llama 3.1 8B and 70B, and Gemini 1.5 Flash and Pro).
>
> See our new version of Figure 6 here: https://anonymous.4open.science/r/bijection-rebuttal-E525/scaling-law.png
>
> While the Llama 3.1 models see a similar trend to Claude and GPT models, we observe a different trend for Gemini that indicates that for Gemini, unlike the other models, attack success is mainly attributed to mismatched generalization. Hence, Gemini serves as an illustrative counterpoint for our analysis on Claude and GPT models. Further discussion below.

---

> ### Author Response · Authors · 2024-12-02
> **Reply to reviewer (2/2)**
>
> **Overall comments**. We agree that we have some burden of proof to show that computational overload is an underlying jailbreak mechanism for bijection learning, not merely a post-hoc artifact of the bijection task. To precisely write down the relevant terms:
>
> *Mismatched generalization*: When prompts are not in plain English, the model reverts to the behavior of the pretrained model, and does not perform classification of unsafe prompts.
>
> *Capability degradation*: The model tries to perform classification of unsafe prompts, but fails because it is not capable enough.
>
> We believe that our experiments collectively provide substantial empirical evidence that for Claude and GPT models, capability degradation is an important mechanism for bijection learning efficacy. In particular:
>
> -----
>
> **I**.   Capability degradation is *sufficient* to explain bijection learning efficacy:
>
> 1.    We show that bijection learning results in capability degradation *beyond* losing functions that are fine-tuned into the model. For example, in Figure 5, we show that difficult bijections cause a model to become *incoherent*, which would not be the case if the model merely reverted to its pre-trained state.
>
> 2.   We show that capability degradation does cause models to fail to classify harmful prompts. In our new experiment (see response to point A above), we showed that weaker models are worse at performing safety classification. In addition, in Figure 8, we show that Claude 3.5 Sonnet fails to classify bijection learning prompts as harmful even when classification instructions are given in English. The model clearly understands, and tries to execute, the classification task, since it correctly labels 100% of benign prompts as safe. However, it rejects just 29% of harmful prompts due to its degraded capabilities (which only results in 12% decrease in ASR in the best-of-6 setting). These results provide evidence that degraded model capabilities are enough to cause jailbreak success.
>
> -----
>
> **II**.   Mismatched generalization is *not sufficient* to explain bijection learning efficacy:
>
> If mismatched generalization were the root cause of bijection learning efficacy, then we should expect bijection learning to achieve maximal success when the target model has fully reverted to its pre-trained state. Thus, peak ASR should coincide with MMLU scores that are around the level of the pre-trained model. Though the MMLU scores of the pre-trained versions of Claude and GPT are unclear, analysis for similar-scale models (PaLM 540B) shows that fine-tuning boosts MMLU accuracy by only 5-10 percentage points (https://www.jmlr.org/papers/volume25/23-0870/23-0870.pdf). The public release fine-tuned models achieve upwards of 90-95% MMLU accuracy (10-shot; see Figs 6 and 11), so we would expect ASR to peak at around 80-90% MMLU accuracy if the underlying mechanism were mismatched generalization. Instead, our ASR peaks at around 55-70% MMLU accuracy, indicating that capabilities must degrade beyond those of the pre-trained model for maximal efficacy.
>
> ----
>
> Gemini 1.5 provides an example of what we expect a mismatched generalization-based Pareto frontier to look like. We get peak ASRs on these models with low-complexity bijection attacks, i.e. easy bijections that produce little-to-no computational overload as evidenced by MMLU evaluations, which makes sense because even bijections with dispersion of 2 or 3 diverge significantly from English plaintext and constitutes a writing task not present in most post-training distributions. However, all other models of similar capability level--i.e. GPT-4o, Claude, and Llama 3.1 models--are not jailbroken by bijections without capability degradation, so mismatched generalization cannot adequately explain the bijection jailbreak for these models.
>
> These models were all released after previous encoding-based jailbreaks gained recognition, and thus could have been more robustly fine-tuned to generalize safety guardrails to encoded user requests. Nevertheless, our results show that *even if safety guardrails completely generalize across all possible prompts, which may be the case for certain models like Claude or GPT for which the developers tried really hard to robustify their safety training, these guardrails can be ineffective for attacks like bijection learning due to degradation of capabilities*.
>
> Overall, while mismatched generalization is clearly at play for most previous work on encoding-based jailbreaks (in particular https://openreview.net/forum?id=MbfAK4s61A, which merely “tricks” the model into believing it is not writing in English), our evidence shows that capability degradation is a prominent mechanism for bijection learning as a jailbreak.
>
> Thanks again for your discussion and we hope our response addresses your questions.

---

> > ### Author Response · Authors · 2024-12-02
> >
> > Dear reviewer, since the last chance for reviewers to ask additional questions is tonight, we would like to inquire about whether our additional experiments and analysis increase your confidence in our work. We value your opinion and would appreciate a chance to answer any further questions or concerns you may have.

---

### Author Response · Authors · 2024-11-22
**Response to all reviewers**

We thank all the reviewers for their thoughtful and helpful insights. We have made significant revisions to the paper based on feedback and encourage all reviewers to revisit the revised version of the paper, especially the updated Table 1 with comparisons to baselines.

Here, we discuss a few shared concerns between reviewers.

**Novelty claims**. We agree that we could do more to differentiate our method from prior work and make the case that our contributions are novel and significant. We rewrite our related works and contributions sections (1.1 to 1.3) and refer reviewers to these sections for further context. However, we also think it’d be useful to summarize our case here for further discussion.

Our attack is “universal,” which means an attacker can use it by inserting their prompt into a template using a simple procedure (encoding the attack intent in bijection language) that does not require any human or AI assistance. Most jailbreaks are not universal. For example, TAP requires a significant feedback loop between an attacker model and the target model that greatly diminishes its ease of use compared to a universal jailbreak.

Previous encoding-based attacks encode a harmful prompt in some particular encoding, like Base64, ASCII, or ROT13 that the model already knows about through training. Compared to these works:
1. We are the first to leverage models’ ability to learn encodings in-context, not just those a model has memorized.
2. Our sampling method is a powerful "fuzzing" technique that allows us to perform best-of-n sampling from an essentially endless pool of possible encodings, which doubles our ASR compared to using only a single random encoding. Best-of-n sampling is not possible using previous techniques – an attacker can keep trying different well-known ciphers, languages, etc. that have been proposed in other papers, but will eventually run out of options.
3. We use quantitative parameters to scale the encoding difficulty to best match the model’s capability level. In particular, our dispersion parameter d, 0<=d<=26, controls how many different letters are mapped to an encoded sequence, where d=0 is plain text and d=26 permutes all letters. Previous attacks are rather blunt: either the model knows the selected encoding or it doesn’t. However, for bijection learning, if a certain model cannot learn to permute all the characters, we can tone down the dispersion to map only a few letters to something else. Hence, our method is uniquely scale adaptive in a way that other methods are not.
4. Our capabilities vs ASR analysis proposes a new reason why our attack works, which is that translating bijections is a hard enough task that it saps the model’s capabilities overall and in particular steals capacity away from safety mechanisms. This is different from the analysis from previous work of “safety tuning is in English, so it doesn’t generalize to other languages” and exposes a potentially new direction in jailbreaking.
5. Finally, our empirical results are interesting, since we achieve extremely high ASRs on frontier models compared to previous methods. See next point.

**Better comparison to baselines**. Reviewer WL23 noted that we should add more encoding baselines. Reviewers CzKw and sNn1 noted that it may not be fair to compare best-of-n to baselines that only create one attack prompt.

Best-of-n sampling is only possible with our method and is one of its main advantages. However, we agree that it would be interesting if we artificially gave previous work this degree of freedom. As suggested, we ensemble 11 of the most popular and best performant encoding baselines (7 more than before) and consider this ensemble attack successful for an attack intent if any one attack succeeds, similar to best-of-11 sampling. **Bijection learning still outperforms by over 30 percentage points on all models** (see Table 1). In fact, best-of-1 bijection learning still outperforms this ensemble on Claude models, and best-of-5 bijection learning outperforms on all models (see Table 3).

Reviewers also suggested we compare to PAIR and TAP, which we add to Table 1. Again, we outperform by over 25 percentage points on all models when using the same (or larger, in the case of TAP) attack budget with these attacks compared to our budget for bijection learning.

**(New) Defense techniques**. We add an analysis of a defense strategy against bijection learning in Section 5: a guard model defense that performs safety classification on model inputs and outputs, nullifying the output when unsafe behavior is detected. The input filter takes the entire context, and the output takes the entire context plus the model response. Curiously, even when the guard model is the same as the target model (Claude 3.5 Sonnet), the ASR of bijection learning is only reduced by 10 percentage points, so even “safety-capability parity” [1] may not be sufficient to defend models from bijection learning.

[1] https://arxiv.org/abs/2307.02483

---

### Meta-Review · Area_Chair_qHhw · 2024-12-13

**Metareview:**

The paper proposes a novel bijection method leveraging side-channel techniques to jailbreak large language models (LLMs). The findings indicate that stronger LLMs are more vulnerable to jailbreak attacks and demonstrate state-of-the-art (SOTA) attack success rates. The proposed method also achieves high success rates in jailbreaking Claude, a leading LLM. These claims are supported by experimental results and comparisons to baseline attack techniques.

The strengths of the paper include its highly effective method for jailbreaking LLMs, which outperforms existing approaches in terms of success rate. The methodology is rigorously evaluated, demonstrating robustness across multiple LLMs, including Claude. This highlights the practical implications and potential risks of deploying strong LLMs in sensitive environments. The paper also finds that that stronger models are easier to jailbreak.

The weaknesses of the paper lie in its lack of novelty and limited scientific contribution to advancing the understanding of LLM vulnerabilities. While the bijection method is effective, the finding does not offer new insights into the fundamental mechanisms underlying these vulnerabilities. This paper would be useful if this capacity degradation argument was real and substantiated. Unfortunately, in the current state, their argument does not seem fully convincing

The primary reason for accepting the paper shall be its empirical results, which stands out as one of the better jailbreak papers. However, there would also be reasons to reject this paper due to its insufficient contribution to advancing scientific understanding. While the method demonstrates strong empirical results, the findings do not significantly enrich the field's knowledge or provide actionable recommendations for addressing the security challenges posed by jailbreaking attacks.

**Additional Comments On Reviewer Discussion:**

All reviewers found the paper to be empirically well done, achieve advantage over most jailbreak papers.

However, in the discussion phase, all reviewers find the paper to be not exciting, and lack new understanding.

---

### Decision · Program_Chairs · 2025-01-22

Accept (Poster)